# i3Deep: Efficient 3D interactive segmentation with the nnU-Net

**Karol Gotkowski**[1,2]                    karol.gotkowski@dkfz-heidelberg.de
**Camila Gonzalez**[3]                      camila.gonzalez@gris.tu-darmstadt.de
**Isabel Kaltenborn**[4]                    IsabelJasmin.Kaltenborn@kgu.de
**Ricarda Fischbach**[4]                    Ricarda.Fischbach@kgu.de
**Andreas Bucher**[4]                       AndreasMichael.Bucher@kgu.de
**Anirban Mukhopadhyay**[3]     anirban.mukhopadhyay@gris.tu-darmstadt.de

[1] *Applied Computer Vision Lab, Helmholtz Imaging*

[2] *Division of Medical Image Computing, German Cancer Research Center (DKFZ), Heidelberg*

[3] *Darmstadt University of Technology, Karolinenpl. 5, 64289 Darmstadt, Germany*

[4] *University Hospital Frankfurt - Institut for Diagnostic and Interventional Radiology, Theodor-Stern Kai 7, 60590 Frankfurt am Main*

## Abstract

3D interactive segmentation is highly relevant in reducing the annotation time for experts. However, current methods often achieve only small segmentation improvements per interaction as lightweight models are a requirement to ensure near-realtime usage. Models with better predictive performance such as the nnU-Net cannot be employed for interactive segmentation due to their high computational demands, which result in long inference times. To solve this issue, we propose the 3D interactive segmentation framework i3Deep. Slices are selected through uncertainty estimation in an offline setting and afterwards corrected by an expert. The slices are then fed to a refinement nnU-Net, which significantly improves the global 3D segmentation from the local corrections. This approach bypasses the issue of long inference times by moving expensive computations into an offline setting that does not include the expert. For three different anatomies, our approach reduces the workload of the expert by 80.3%, while significantly improving the Dice by up to 39.5%, outperforming other state-of-the-art methods by a clear margin. Even on out-of-distribution data i3Deep is able to improve the segmentation by 19.3%.

**Keywords:** interactive segmentation, nnU-Net, uncertainty, out-of-distribution

## 1. Introduction

Manual segmentation of 3D medical data such as CT, MRI or ultrasound scans is highly time-consuming, as it often consists of hundreds of slices. Interactive segmentation reduces the workload on experts by refining the segmentation from user interactions with the goal to minimize the necessary amount and thus saving the expert time. Such methods could enable an expert to segment a CT scan with just a few clicks.

The two requirements for interactive applications are a **high predictive performance** and a **low reaction time** ($< 1s$). The first enables the expert to annotate the image with much fewer interactions than when done manually, while the latter ensures the application is usable in practice. Current approaches limit the model capabilities as all their computations are performed live. To this day, no approaches exist to our knowledge that try to lift

this limitation and benefit from the much higher predictive performance of larger models. Our framework addresses this and provides an alternative by moving the expensive computations into an offline setting. Not only does this lead to fast reaction times, but also enables the use of large models, which provide much better segmentation results. Our method consists of the following steps, illustrated in Figure 1.

First, we extract both initial segmentations and uncertainties with a presegmentation nnU-Net for a subject. Based on the uncertainties, we automatically select a small number of slices with a **one-shot slice acquisition function** and send these to the expert for corrections. The corrections are then used by a **refinement nnU-Net** to improve the segmentation globally by inferring from the local corrections. Both the presegmentation and refinement nnU-Nets are trained once beforehand, with the framework solely relying on inference during the interactive segmentation process.

The expert is not involved in the presegmentation or refinement stages, which reduces the practical reaction time for the framework to zero. As a one-shot slice acquisition function is used, only a single iteration with the framework is needed to significantly improve the segmentations.

We demonstrate the effectiveness of our approach with an evaluation on the brain tumor and pancreas datasets from the Medical Segmentation Decathlon and an **out-of-distribution** in-house chest CT scan dataset with COVID-19 lesions. The code is open source and released at: https://github.com/Karol-G/i3Deep

## 2. Related Work

A number of interactive segmentation approaches have been proposed over the years, which we discuss in the following. An overview of the relevant methods in regards to predictive performance and reaction time is given in Table 1.

|  | Classical | U-Net/FCN | Konyushkova | P-Net/iW-Net | i3Deep |
|---|---|---|---|---|---|
| Predictive Performance | Low | Medium | Low | Medium | High |
| Reaction Time | Medium | Slow | Fast | Fast | Instantly |

Table 1: Predictive performance and reaction times of interactive segmentation approaches.

**Classical methods** that are still popular today in the medical domain are Graph-Cut (Greig et al., 1989), Watershed (Meyer, 1994) and Random Walker (Grady, 2006). These methods are relatively fast even on 3D data, but have a low predictive performance by current standards.

Deep interactive segmentation approaches often outperform classical methods and most of them follow a very similar pattern of pretraining a refinement model with simulated user input and then running inference with actual expert input. However, processing higher resolution 3D images is computationally very expensive with CNNs. Therefore, approaches that employ a **U-Net** or **FCN** have slow reaction times as it is the case with Bredell et al. (2018); Li et al. (2021) and the 3D Slicer implementation of Sakinis et al. (2019).

As an alternative, other approaches use very lightweight 3D models like the **P-Net** (Wang et al., 2018, 2019b; Lei et al., 2019; Liao et al., 2020; Xu et al., 2021) or **iW-Net** (Aresta et al., 2019), which achieve a near-realtime reaction time, but have a lower predictive

performance in turn.

Besides the approaches that are task-agnostic, there are also a number of methods that are tailored to specific tasks like prostate, cell or vessel segmentation (Cheng and Liu, 2017; Koohbanani et al., 2020; Dang et al., 2022).

Other approaches used in active learning, such as by Konyushkova et al. (2015, 2019), use Boosted Trees uncertainties to find areas that should be corrected by an expert. Drawbacks of this method are the limited predictive performance and the need to retrain after every iteration.

## 3. Methodology

The i3Deep framework uses the nnU-Net (Isensee et al., 2021) for both the presegmentation and refinement model, as it has a very high predictive performance and achieves state-of-the-art results on many medical benchmarks. The training process of both models is explained in 3.1 and the inference pipeline of i3Deep is outlined in 3.2.

### 3.1. Presegmentation & refinement nnU-Net training

We presume that a small number of subjects is already annotated, which make up the train set. Both the presegmentation and the refinement nnU-Net are trained exclusively on this train set once. The presegmentation nnU-Net is trained in a normal fashion, while the refinement nnU-Net further uses the ground truth annotations of the training set to simulate user interactions. For each image during training, slices of the ground truth are randomly chosen and all other slices are set to zero in the image volume. This modified image volume is then concatenated along the channel dimension of the image data and used as training input. When presented with corrected slices during inference, the refinement model is then able to utilize the corrections.

### 3.2. Inference pipeline

The inference pipeline consists of a four-stage process depicted in Figure 1.

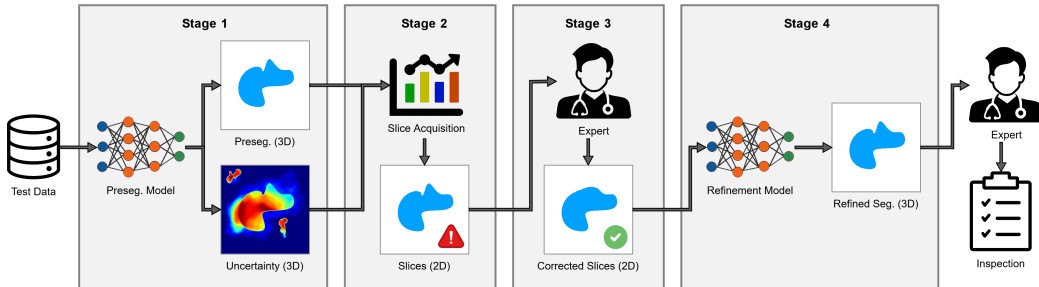

Figure 1: Overview of our proposed i3Deep framework and its four stages.

### 3.2.1. Stage 1: Presegmentation & uncertainty computation

In stage one, the presegmentation model is used to run inference on new unseen subjects to provide presegmentations alongside uncertainties from the model. Estimating the uncer-

tainties for our approach can be done with multiple uncertainty predictors such as Test-Time Augmentation (Wang et al., 2019a), Monte Carlo Dropout (Gal and Ghahramani, 2016) or Deep Ensembles (Lakshminarayanan et al., 2017), which provide multiple varying predictions for an image. The voxel uncertainty inherent to the variations of these predictions is then quantified by computing their entropy. The uncertainty estimation process is expanded on in Appendix A.

### 3.2.2. Stage 2: Slice acquisition

In stage two, a one-shot slice acquisition function selects multiple slices for each subject in axial, coronal and sagittal orientation from the 3D image based on the quantified uncertainties. The goal of this acquisition function is to select the minimum number of slices necessary to maximally improve the segmentation in a single run.

First, for each slice the sum of all uncertainty voxels is computed. Next, slices that have less uncertainty than any other slice within a minimum distance $minDist$ are removed. This leaves only slices that are local maxima and decreases uncertainty correlation between slices. Afterwards, slices that have not enough uncertainty are removed as well, based on a $minUncert$ parameter. Of the remaining slices, further, only a subset of $maxSlices$ is selected that have the highest uncertainty. All three parameters are optimized after the training of the presegmentation nnU-Net once on validation data.

### 3.2.3. Stage 3: Expert annotations

In this stage, the expert is involved in the process for the first time. The acquired slices of the previous stage are sent to the expert for correction. The expert is provided for each slice the presegmentation and subsequently corrects any mistakes they identify. We opt to let the expert choose their preferred annotation tool to enable precise corrections even on images with diffuse class borders, as it is the case with COVID-19 lesions. It is important to note that stage 1 and 2 both happen in an offline setting and the expert is only involved once these stages have been completed.

### 3.2.4. Stage 4: Refinement

In stage four, the refinement model is used to improve upon the segmentation as depicted in Figure 2. The corrected slices are projected into an empty volume back into their original

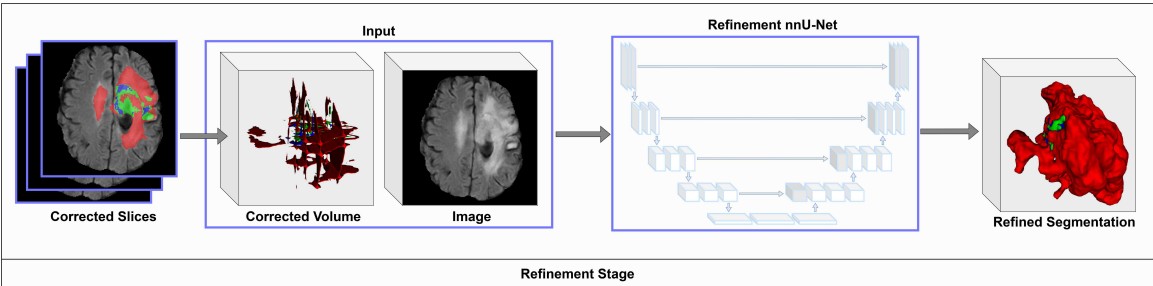

Figure 2: Inference process of the refinement nnU-Net with the corrected slices.

positions. Then this volume is concatenated with the original image and used for inference by the refinement model, which significantly improves the segmentation.

## 4. Experimental Setup

### 4.1. Datasets

We evaluate on three datasets to prove the applicability of our approach to a number of use cases. First, we use the Medical Segmentation Decathlon (MSD) **brain tumor** (Antonelli et al., 2021) dataset consisting of 484 labeled brain MRI scans with 5 MRI-modalities. The labeled classes are *edema*, *non-enhancing tumor* and *enhancing tumor* and the mean subject size of the dataset is 240x240x155 voxels. We split the dataset into a train set of 100 subjects, a validation set of 50 subjects and a test set of 334 subjects.

The MSD **pancreas** (Antonelli et al., 2021) dataset consists of 281 labeled portal-venous phase CT scans with the classes *pancreas* and *cancer* and a mean subject size of 512x512x98 voxels. Again, we split the dataset into a train set of 100 subjects, a validation set of 36 subjects and a test set of 145 subjects.

The third dataset is a **COVID-19** dataset, which consists of COVID-19 chest CT scans with the label *ground-glass opacity* (GGO). The dataset is divided into a set of subjects that are publicly available (MedSeg; Jun et al., 2020; Morozov et al., 2020) and an out-of-distribution (OOD) in-house private set to evaluate the generalizability of our approach. In total, the dataset consists of 129 subjects and a mean subject size of 1280x1280x266 voxels. The data is split into a train set of 79 subjects, a validation set of 10 subjects and an in-house OOD test set of 40 subjects.

### 4.2. Baselines

We compare our approach to other state-of-the-art 3D interactive segmentation techniques that focus on fast reaction times for the expert and can thus be used in practice. Approaches that have long reaction times such as Bredell et al. (2018); Li et al. (2021); Sakinis et al. (2019) are excluded due to their missing practicality. For the classical methods, we compare against Graph-Cut (Jirik et al., 2018; Jirik), Watershed (Skimage) and Random Walker (Skimage). For CNN-based methods, we compare against the P-Net from DeepI-GeoS (Wang et al., 2019b), which is used in most fast CNN-based approaches. We found during training that the used geodesic distance transforms from DeepIGeoS drastically decrease the performance in our setting and thus opted train the P-Net in the same fashion as our refinement nnU-Net instead. Further, to be able to fairly compare all baselines, they all receive the exact same corrected slices as the refinement nnU-Net from i3Deep.

### 4.3. Training details

Training of the presegmentation and refinement nnU-Nets was done in PyTorch with SGD optimizer, a learning rate of 1e−2, a weight decay of 3e−5, a momentum of 0.99 and 1000 epochs of training time. The P-Net used the same settings, but with grid-search optimized learning rates for the brain tumor, pancreas and COVID-19 datasets of 1e−2, 1e−4 and 1e−4, respectively. The parameters of the acquisition function were optimized to a $minDist$ of 0.0234, $maxSlices$ of 12 and $minUncert$ of 0.1.

## 5. Results

### 5.1. Predictive performance

We conduct an evaluation of the predictive performance in terms of Dice score performance over all datasets. The results are shown in Figure 3 and as table in Appendix B.1. Based on our uncertainty ablation study in section 5.3, we choose Deep Ensembles as the used uncertainty predictor for the presegmentation nnU-Net. However, other uncertainty methods can be used as well and are viable options for i3Deep.

Starting with the brain tumor dataset (red plots in Figure 3), we can see that the presegmentation (blue) performs acceptable for the classes *edema* and *enhancing tumor*, but rather bad for *non-enhancing tumor*. By contrast, i3Deep with nnU-Net refinement (orange) outperforms the presegmentation and all other baselines over all classes by a margin of up to 19.2%. Compared to the presegmentation, i3Deep improves the mean Dice score by 8.1%, 19.2% and 7.2%, respectively. The improvements for edema and non-enhancing tumor are lower as the Dice scores are already high for the presegmentation and thus only limited improvements are possible.

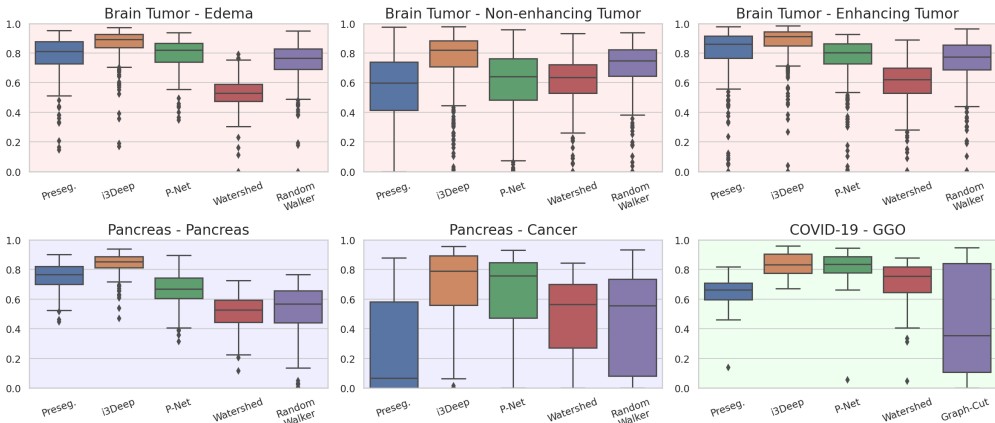

Figure 3: Box plots for different classes of the brain tumor, pancreas and COVID-19 dataset for the presegmentation, our method and all baselines.

Next, we inspect the results for the pancreas dataset (blue plots in Figure 3). For the *pancreas* class, we see again a significant improvement with i3Deep in comparison to the presegmentation by 8.4%. By contrast, all baseline methods perform significantly worse than the presegmentation, which shows their limited predictive performance. For the *cancer* class, we see that the presegmentation fails completely with a mean Dice score of only 27.4% due to the difficulty of separating the small cancer class from the pancreas class, by which it is often surrounded. In this instance, i3Deep manages to improve the segmentation by a margin of 39.5%. The P-Net improves the Dice score by 34.1%, which is also considerable. Yet, it shows again the predictive limitations of this lightweight model. The other baselines manage to improve the Dice score, but are significantly worse than i3Deep and the P-Net.

The last dataset we evaluate is the COVID-19 test set (green plot in Figure 3). It is important to note, that i3Deep has never seen any of our in-house data during training, thus

making the test data out-of-distribution (OOD) and an important benchmark for the practical usability of i3Deep. Here, the presegmentation achieves a Dice score of 64.4%, which is acceptable for OOD data. However, even though the data is OOD i3Deep still improves the Dice score by 19.3% to 83.7%, showing the applicability of our approach for real world usage. This time, the P-Net achieves a similar performance with a Dice score of 80.09%. The other baselines are again considerably worse, with Graph-Cut showing even a very high variance in terms of predictive performance.

In summary, i3Deep can improve the segmentation quality significantly in comparison to state-of-the-art baselines, while enabling the usage of models with high predictive performance such as the nnU-Net in an interactive setting.

## 5.2. Qualitative comparison

In Figure 4 a qualitative comparison of the brain tumor dataset is shown. Here, the presegmentation model fails to detect a part of the non-enhancing tumor (green) and only badly predicted the enhancing tumor (blue). By contrast, i3Deep manages to recover the missing regions almost perfectly with only minor inaccuracies for the enhancing tumor. The P-Net also recoveres some of the regions, but the overall prediction lacks the same quality as that of i3Deep. The predictions for Watershed and Random Walker also recover small amounts of the missing regions, but are worse in comparison to both i3Deep and the P-Net. The pancreas and COVID-19 dataset comparison (Appendix B.2) further confirm our results.

In conclusion, all refinement models managed to recover missing lesions, yet i3Deep is the model that achieves the best segmentation in comparison to the ground truth. This shows the importance of using models with a high predictive performance in interactive settings to reliably provide segmentations of high quality for the expert.

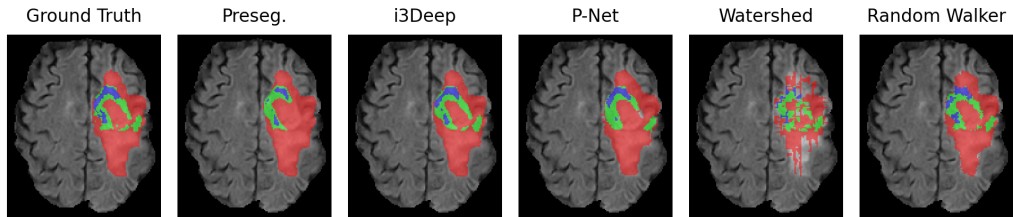

Figure 4: Qualitative comparison of the ground truth, the presegmentation, our approach and the baselines on the brain tumor dataset.

## 5.3. Uncertainty ablation

We conduct an ablation study to determine the uncertainty predictor for the presegmentation model that performs best with our approach. It is important to note that the tested uncertainty predictors are only used for the presegmentation model, as the refinement model does not need to compute uncertainties. In total, we compare the predictors Test-Time Augmentation (TTA), Monte Carlo Dropout (MC Dropout) and Deep Ensembles. The evaluation is done on all three validation datasets and measured in terms of Dice score. The results are shown in Figure 5. The Dice scores show that all predictors perform

quite similar on the brain tumor dataset with Deep Ensembles being only 0.8%, 1.3% and 0.1% better in the mean than the second best predictor on each class respectively. On the pancreas dataset the results are clearer with Deep Ensembles surpassing the second best predictor in the mean by 2.4% and 5.1%, respectively. However, Deep Ensembles perform 1.5% worse than TTA on the COVID-19 dataset. As Deep Ensembles have the best performance on most classes, we choose it as our predictor for our evaluation in section 5.1. Yet, the evaluation also shows that all three predictors are viable methods.

Further, we evaluate the predictors in terms of ECE for which the results are discussed in Appendix C.1 and reflect these results. We also evaluate the impact of using P-Net Deep Ensemble uncertainties instead of nnU-Net Deep Ensemble uncertainties in Appendix C.2. The results show that the uncertainties of both models are equally good for our approach.

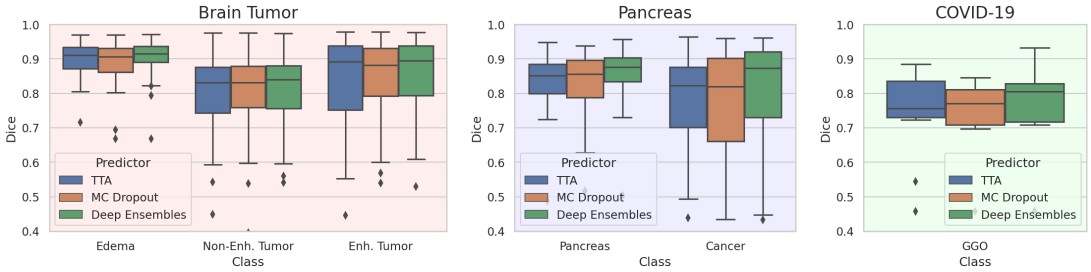

Figure 5: Comparison of the uncertainty predictors TTA, MC Dropout and Deep Ensembles on the brain tumor, pancreas and COVID-19 dataset.

## 5.4. Annotation Ratio

To assess the expected workload reduction we propose the subject-wise *Annotation Ratio* (AR), which measures how many fewer slices need to be annotated: $AR = \frac{|S|}{|GT_{foreground}|}$. Here, $|S|$ denotes the number of all selected slices and $|GT_{foreground}|$ the number of axial ground truth slices that contain foreground annotations.

On the brain tumor dataset we achieve an AR of 20.56%, on the pancreas dataset 17.94% and on the COVID-19 dataset 20.50%. Averaged over all datasets, we achieve an AR of 19.67% meaning that an expert needs to annotate 80.33% less slices of what they would normally annotate, resulting in a significant workload reduction.

## 6. Conclusion

We introduce the interactive framework i3Deep, which enables the usage of models with a high predictive performance. i3Deep provides an expert pre-acquired slices based on uncertainties and uses the expert corrections to improve the segmentation with a refinement nnU-Net. The evaluation shows that this approach reduces the workload of the expert by 80.3%, while significantly improving the segmentations up to 39.5% and outperforming other state-of-the-art interactive methods often considerably. Even on out-of-distribution data, i3Deep is able to improve the segmentation by 19.3%. In the future, we intend to move from slices to patches and evaluate i3Deep in multiple user studies on even more anatomies and out-of-distribution datasets.

## Acknowledgments

Part of this work was funded by Helmholtz Imaging (HI), a platform of the Helmholtz Incubator on Information and Data Science.

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

## Appendix A. Uncertainty estimation

Uncertainty can be estimated by multiple means and the estimation consists of two steps. First, multiple predictions need to be inferred stochastically with methods such as Test-Time Augmentation (Wang et al., 2019a), Monte Carlo Dropout (Gal and Ghahramani, 2016) or Deep Ensembles (Lakshminarayanan et al., 2017), which we refer to as *uncertainty predictors*. Second, the uncertainty from the predictions must be quantified with methods such as the entropy, variance or the bhattacharyya coefficient (Kang and Wildes, 2015), which we refer to as *uncertainty quantification*. We determine the best predictor in section 5.3, but choose entropy for the quantification as it is the most popular one and the influence of the quantification method is limited.

In this context, the *entropy* is defined as the entropy of each voxel belonging to a certain class and is based on the average of multiple predictions. Further, the entropy is divided by its information length to be within the interval of [0,1].

For an image $x$ with $C$ classes and a total of $T$ different predictions $p_{t,c}(x)$ for each class, the entropy is defined as:

$$\overline{p_{T,c}(x)} = \frac{1}{T}\sum_{t=1}^{T} p_{t,c}(x) \tag{1}$$

$$H(p_{T,C}(x)) = \frac{-\sum_{c=1}^{C}\overline{p_{T,c}(x)} * log(\overline{p_{T,c}(x)})}{log(C)} \tag{2}$$

## Appendix B. Results

### B.1. Predictive performance

In this section we report the mean and standard deviation for our results of the brain tumor dataset in Table 2, the pancreas dataset in Table 3 and the COVID-19 dataset in Table 4. Dice scores marked with $*$ denote a $p$-value $< 0.05$ when compared with the second place method. The results are the same as the one depicted in Figure 3.

| Brain Tumor | | | | | |
|---|---|---|---|---|---|
| | Preseg. | i3Deep | P-Net | Watershed | Random Walker |
| Edema | 0.784±0.128 | **0.865±0.103**∗ | 0.792±0.101 | 0.53±0.102 | 0.75±0.124 |
| Non-E. T. | 0.566±0.233 | **0.758±0.192**∗ | 0.596±0.218 | 0.603±0.174 | 0.7±0.182 |
| Enh. T. | 0.792±0.201 | **0.864±0.158**∗ | 0.751±0.186 | 0.598±0.155 | 0.74±0.175 |

Table 2: Mean and standard deviation Dice scores for the edema, non-enhancing tumor and enhancing tumor class of the brain tumor dataset for the presegmentation, our method and all baselines.

| Pancreas | | | | | |
|---|---|---|---|---|---|
| | Preseg. | i3Deep | P-Net | Watershed | Random Walker |
| Pancreas | 0.749±0.096 | **0.834±0.08**∗ | 0.66±0.114 | 0.509±0.116 | 0.525±0.181 |
| Cancer | 0.274±0.309 | **0.669±0.298**∗ | 0.615±0.308 | 0.478±0.274 | 0.467±0.312 |

Table 3: Mean and standard deviation Dice scores for the pancreas and cancer class of the pancreas dataset for the presegmentation, our method and all baselines.

| COVID-19 | | | | | |
|---|---|---|---|---|---|
| | Preseg. | i3Deep | P-Net | Watershed | Graph-Cut |
| GGO | 0.644±0.125 | **0.837±0.079** | 0.809±0.136 | 0.702±0.172 | 0.464±0.357 |

Table 4: Mean and standard deviation Dice scores for the GGO class of the COVID-19 dataset for the presegmentation, our method and all baselines.

## B.2. Qualitative comparison

We continue the qualitative comparison of the pancreas and COVID-19 dataset in this section. Figure 6 shows a comparison for the pancreas dataset.

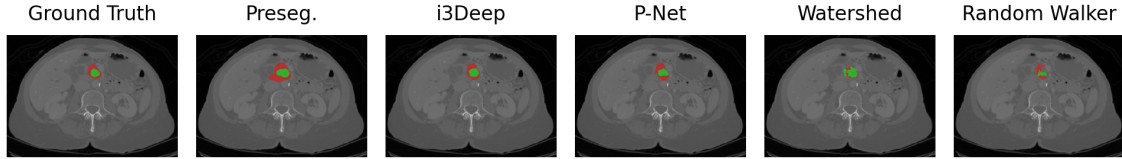

Figure 6: Qualitative comparison of the ground truth, the presegmentation, our approach and the baselines on the pancreas dataset.

Both the pancreas and the cancer class are relatively small with the pancreas class surrounding the cancer class in most subjects. It can be seen, that the presegmentation overestimated both classes. By comparison, i3Deep and the P-Net both reduced this oversegmentation,

yet i3Deep aligned the class borders overall better with the ground truth borders than the P-Net. For Watershed and Random Walker the issue of oversegmentation only increased with either the pancreas or cancer class oversegmenting the entire lesion. The comparison for the COVID-19 dataset is shown in Figure 7. Here, we see that the presegmentation missed the GGO lesions in the lower lungs, while all interactive methods were able to recover the missing lesions. However, we see again similar results with i3Deep being the most precise by not falsely segmenting the sparse small pockets of lesion free lung. The P-Net also recovered the GGO lesions for the lower lungs, but oversegmented the lung in general by segmenting the small lesion free pockets too. Again, the classical methods did not achieve the same level of refinement as i3Deep as both of them are too coarse.

| Ground Truth | Preseg. | i3Deep | P-Net | Watershed | GraphCut |

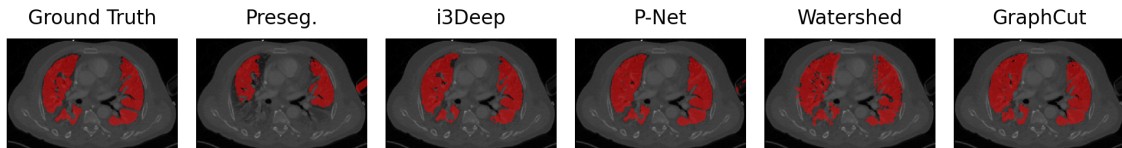

Figure 7: Qualitative comparison of the ground truth, the presegmentation, our approach and the baselines on the COVID-19 dataset.

Similar to our qualitative evaluation of the brain tumor dataset, all refinement models managed to recover missing lesion. However, i3Deep is the method that achieved the best segmentation in comparison to the ground truth, showing the importance of using models with a high predictive performance in interactive settings to reliably provide segmentations of high quality.

## Appendix C. Uncertainty ablation

### C.1. Expected Calibration Error

A common method to determine the quality of uncertainty is the *Expected Calibration Error* (ECE) (Naeini et al., 2015; Guo et al., 2017). The ECE measures the difference in expectation between confidence and accuracy to determine the miscalibration and thus the quality of the uncertainty. It divides the softmax output range of [0,1] into $M$ multiple bins $B_m$ of equal size and measures the accuracy and confidence of the softmax outputs that fall within each bin. A weighted average over the total number of predictions $n$ is taken to compute a scalar miscalibration value. The ECE is formally defined as:

$$ECE = \sum_{m=1}^{M} \frac{|B_m|}{n} |acc(B_m) - conf(B_m)| \tag{3}$$

The accuracy and confidence of bin $B_m$ are defined as follows:

$$acc(B_m) = \frac{1}{|B_m|} \sum_{i \in B_m} 1(\hat{y}_i = y_i) \tag{4}$$

$$conf(B_m) = \frac{1}{|B_m|} \sum_{i \in B_m} \hat{p}_i \tag{5}$$

Here, $\hat{y}_i$ and $y_i$ denotes the predicted class and ground truth class for a prediction $i$ and $\hat{p}_i$ denotes the confidence for a prediction $i$.

Based on this, we evaluated the predictors TTA, MC Dropout and Deep Ensembles on the validation sets of the brain tumor, pancreas and COVID-19 dataset with the ECE measure. The results are shown in Table 5. For the brain tumor and pancreas dataset Deep Ensembles achieve the lowest calibration error and have thus the best uncertainties. By contrast, TTA performs best on the COVID-19 dataset with Deep Ensembles being slightly worse than MC Dropout. We can conclude that Deep Ensembles have probably a slight advantage over the other predictors, yet it is difficult to estimate based on the ECE how relevant that advantage is. However, in conjunction with our Dice score evaluation in section 5.3 we can conclude that this advantage is noticeable but not too significant.

Another important aspect to note is that the calibration seems to be very good based on the ECE results. However, this is most likely only partly the case as the ECE has a number of issues that are discussed in (Nixon et al., 2019) and which are especially true for 3D data, which suffers from severe class imbalance. Still, the ECE is a commonly used measure, hence us including it, but the result should always be taken with a grain of salt. For this reason, our uncertainty evaluation based on Dice score performance is more reliable.

|                 | Brain Tumor | Pancreas | COVID-19 |
| --------------- | ----------- | -------- | -------- |
| Deep Ensembles  | 0.0008      | 0.0004   | 0.0103   |
| MC Dropout      | 0.0012      | 0.0011   | 0.01     |
| TTA             | 0.0013      | 0.0006   | 0.0086   |

Table 5: The ECE results for the uncertainty predictors TTA. MC Dropout and Deep Ensembles over all three datasets.

### C.2. nnU-Net & P-Net uncertainty comparison

We evaluated the impact of using a different underlying model when using Deep Ensembles for the uncertainty computation. For this purpose, we computed uncertainties with a nnU-Net and a P-Net Deep Ensemble on the brain tumor dataset and used the resulting uncertainties to compare the predictive performance of the refinement nnU-Net and refinement P-Net. The mean Dice score result are shown in Table 6.

The performance of both nnU-Net refinement models is almost the same and independent of the uncertainty generating underlying model. The results for the refinement P-Net are similar with no significant change in model performance. However, due to the fact that the nnU-Net has a considerably better predictive performance, the refinement nnU-Net achieves a significantly better mean Dice score across all classes than the refinement P-Net.

We can conclude that there is no impact of using a different presegmentation model for uncertainty computation in our setting when using Deep Ensembles.

| Brain Tumor | | | |
|---|---|---|---|
| | i3Deep (nnU-Net U.) | i3Deep (P-Net U.) | P-Net (nnU-Net U.) | P-Net (P-Net U.) |
| Edema | **0.865±0.103** | 0.863±0.105 | 0.792±0.101 | 0.785±0.116 |
| Non-E. T. | 0.758±0.192 | **0.770±0.183** | 0.596±0.218 | 0.615±0.204 |
| Enh. T. | **0.864±0.158** | 0.863±0.1655 | 0.751±0.186 | 0.75±0.188 |

Table 6: Mean and standard deviation Dice scores on the brain tumor dataset for the i3Deep nnU-Net refinement model and P-Net refinement model evaluated with nnU-Net and P-Net uncertainties. The term *Uncertainties* has been denoted as *U*.

