# OpenReview forum: "i3Deep: Efficient 3D interactive segmentation with the nnU-Net"
_MIDL.io/2022/Conference — MIDL 2022_

### Official Review · Reviewer_V4xw · 2022-01-24

**Confidence:** 4
**Preliminary Rating:** 4
**Recommendation:** Poster

**Summary:**

This paper introduces a novel human-in-the-loop method for annotation of 3D medical images. The framework is based on the state-of-the-art nnU-net method and the main contribution is the adaptation to adding a human annotater in the pipeline. The framework is validated on several larger medical data set with convincing results.

**Strengths:**

The paper demonstrates and validates a novel human-in-the-loop framework. Having a human annotater in the pipeline when annotating medical images has shown to be crucial in many applications. With this article the authors demonstrates that their approach both makes annotation faster and more accurate in terms of DICE scores.


**Weaknesses:**

The main weakness is the amount of novelty in the approach. While the methodological framework seems sound it still is a smaller addon to the already impressive nnU-net.
Some text could be rearranged to make the paper more interesting. For example adding text on uncertainty estimation in the main body of text.

**Deanonymize Review:**

no

**Detailed Comments:**

The paper is well structured and easy to read. It is easy to understand the overall topic, the motivation, the idea and the results. It also seems doable to redo the experiments.
The authors have chosen to place quite a lot of material in the appendix. I would have added some more of the uncertainty estimation in the main text. Some text could be removed. For example the introduction paragraph in section 4 and 5 just gives an overview of the coming text and are not really needed for such a short text.


**Final Rating After The Rebuttal:**

4: Weak Accept

**Justification Of The Final Rating:**

I thank the authors for addressing the comments on the paper and I can also see that the paper has improved. In general, I am happy with the paper and I am curious on how this approach to active learning will evolve. I am keeping my recommendation on a weak accept.

**Paper Type:**

validation/application paper

**Questions To Address In The Rebuttal:**

In general it is an easy to read manuscript. I would suggest cutting some text in the main text like the introduction in section 4 and 5. Perhaps also moving the algorithm 1 to an appendix. Then I would add some more text on the uncertainty estimation in the main text.

**Special Issue:**

no

---

### Official Review · Reviewer_t12B · 2022-01-24

**Confidence:** 5
**Preliminary Rating:** 3
**Recommendation:** Poster

**Summary:**

This paper proposes an interactive workflow for 3d segmentation of medical images. It uses nnUNet [Isensee et al. 2021] multi paradigm segmentation system together with Deep Ensembles [Lakshminarayana et al. 2017] to identify uncertain predictions. A minimum set of 2d planes is selected and presented for the user to correct. The training and updating is organized in an off-line fashion to improve effective human-computer interaction. The method is tested on 3 data sets: 2 from the Medical Segmentation Decathlon [Antonelli et al., 2021], brain tumor with 484 labelled MRI scans in 5 modalities, and pancreas with 281 labeled CT scans, and 1 from COVID19 CT dataset, the origin of which is unclear. The method is compared with PNet [Wang et al., 2019], Graph-cut, watersheds and random walker. All methods are trained on the improved annotations. and the presented method outperforms them all.

**Strengths:**

The paper is well written and presents a useful framework for giving experts of limited time the ability to give input to slow but powerful state-of-the-art segmentation methods. The paper has a thorough investigation into the optimal uncertainty prediction method, and it compares with other even older methods from the literature. The application to the 3 different and recent publicly available datasets is convincing.

**Weaknesses:**

While I very much appreciate the work, it is unclear to me, what is being tested. The expert time per segmentation improvement is optimized, but this strategy is not compared with other active-learning methods. Further, although the segmentation results are impressive, this seems to be the result of nnUNet and not the proposed method, since all methods are trained on the same data. Finally, while it is nice to stroll down memory lane, it seems necessary to include Watersheds, and possibly one of the other old methods could also be replaced with a more up-to-date method not part of the nnUNet-framework.

**Deanonymize Review:**

yes

**Detailed Comments:**

I have no detailed comments.

**Paper Type:**

methodological development

**Questions To Address In The Rebuttal:**

The focus of this paper is confusing. It contains much careful and well documented work, but it seems to fail its goal: I would like the authors to argue what the added value of their interactive method is as compared to other active learning methods.

**Special Issue:**

no

---

### Official Review · Reviewer_7mjc · 2022-01-28

**Confidence:** 4
**Preliminary Rating:** 1
**Recommendation:** Poster

**Summary:**

The authors propose a framework, i3Deep, for interactive segmentation, where a presegmentation model together with a refinement model are used to leverage user corrections applied to selected slices from 3D volumes to improve the refinement model performance. The proposed approach uses uncertainty to select slices for dense segmentation which would most benefit the model. The authors use a nnU-net model in the proposed approach, and compare to a P-net model as well as the Watershed and Random Walker algorithms. The proposed approach out-performs these baseline methods, shown quantitatively and qualitatively.

**Strengths:**

The authors propose the use of uncertainty measures to identify slices which will benefit the refinement model, which would seem especially helpful with 3D medical image data as it can be time-consuming to inspect all relevant regions in the image.

The authors perform their analysis with multiple open-source datasets, including the MSD brain tumour, MSD pancreas, and a COVID-19 dataset, as well as an out-of-distribution in-house dataset to evaluate generalizability of the proposed method.

The authors provide both qualitative and quantitative results across these datasets, and include additional analyses comparing the performance of the framework using different ways of computing uncertainty (deep ensembles, test-time augmentation and MC dropout).

**Weaknesses:**

There is limited novelty in the paper, with the main contribution being the use of uncertainty to select slices for user correction in order to improve performance over a baseline segmentation network. However, it is not clearly explained how the uncertainty is used, as detailed in the next section "Detailed Comments".

The main motivation for the proposed method appears to be that the authors want to use "larger models ... such as the nnU-Net" for their "high predictive performance" while also achieving a "low reaction time". The authors claim that "approaches that employ a U-Net or FCN have slow reaction times as it is the case with Bredell et al. [sic]; Li et al. (2021) and the 3D Slicer implementation of Sakinis et al. (2019)", and  "Until now, research has focused on fast lightweight models, as otherwise the inference times are too long for larger models that have a high predictive performance such as the nnU-Net (Isensee, 2021)." The authors have not defined what a suitably "low reaction time" is, but it is difficult to understand this motivation when in fact the first cited work (Bredell et al.), which uses U-nets in an interactive, iterative fashion, required only 3.9ms for updated model predictions per interactive iteration.

The authors also overlook the required time to produce the type of correction used in the paper compared to previous approaches - namely the authors propose the use of dense segmentations of multiple slices from 3D images, whereas all cited approaches use minimal user interaction in the form of clicks, scribbles and/or bounding boxes (Bredell et al.), (Wang et al., 2019b), (Sakinis et al., 2019), (Li et al., 2021).

Aside from the non-CNN methods considered, the authors essentially compare a nnU-Net model with a lighter-weight P-Net model, using the selected slices resulting from the nnU-Net model uncertainty and dense annotations for corrections to the refinement model for both. This is not a meaningful comparison, as (a) the P-net model was not initially proposed for use with dense segmentation corrections in (Wang et al., 2019b), and (b) one would expect the larger nnU-Net model to perform better when they are both trained in the same manner with dense segmentations for the refinement model.

The methodology could be more clearly explained, particularly around the use of uncertainty for slice selection, exactly how the refinement model is trained, and how hyper-parameters were selected for the baseline P-Net model.

**Deanonymize Review:**

no

**Detailed Comments:**

The use of uncertainty to select slices for iterative training is very intuitive, but the description of how this is achieved is extremely difficult to understand. The authors do not provide an explanation of the approach in the text of "Section 3.2. Stage 2: Slice acquisition", but instead provide pseudo-code in "Algorithm 1". Unfortunately, this pseudo-code is very difficult to interpret, containing terms which are not defined (e.g. "pos", "s", "l", the hat symbol), as well as terms which are left up to the reader's interpretation (e.g. "Length(s)", "SubFromEach(s, pos)", "uncertainSlices = Append(uncertainSlices, Sum(slice))"). The  "while" condition is particularly difficult to understand, and appears to be the crux of how slices are selected based on uncertainty.

There are several crucial details of the comparison to baselines which are unclear or misleading. Firstly, the authors compare to one other CNN-based approach, DeepIGeoS (Wang et al., 2019b). They explain that they opted to train the P-Net used in DeepIGeoS in the same fashion as their refinement nnU-net instead of using the geodesic distance transforms proposed in (Wang et al., 2019b), since the latter led to significantly worse performance. The approach proposed in (Wang et al., 2019b) used a fundamentally different form of user interaction to what the authors propose - namely in (Wang et al., 2019b), users are expected to perform minimal edits by providing only scribbles or clicks, thereby minimizing user-interaction time while improving model performance. This is in fact the same motivation and expected interaction as in other cited works which use CNNs, namely (Bredell et al.), (Li et al., 2021) and (Sakinis et al., 2019). The authors propose that multiple entire slices be corrected with dense segmentations in the current work, which would require orders of magnitude (10s of minutes compared to several seconds) more time to produce than scribbles or clicks. For a fair assessment when comparing two approaches which use entirely different user-interactions, the authors should also compare the effect on annotation time.

Secondly, given that the method in (Wang et al., 2019b) is not directly compared to because of this fundamental difference in user-interaction, the P-net appears to be trained in the same way as the nnU-Net within the proposed framework. No details are provided about the differences in training of the P-net model(s) to the nnU-Net model(s) - are both a presegmentation model and a refinement model trained using a P-net architecture in the former case? Are hyper-parameters the same for training both models? If both of these are the case, then it would be entirely unsurprising that the nnU-Net model outperforms the lower capacity P-net model. This difference would also only be enhanced by the choice to use slice selection for all baselines based on uncertainty of the nnU-Net model (for example, the worst performing slices for the presegmentation nnU-Net may not coincide with the worst performing slices of a presegementation P-net).



**Final Rating After The Rebuttal:**

3: Borderline

**Justification Of The Final Rating:**

I thank the authors for a very detailed set of responses. The authors have addressed most of my initial concerns, and have significantly improved the clarity of the proposed approach. Specifically, it is now clear how the refinement model is only trained once using randomly selected slices with GT annotations concatenated with the input image. This paradigm allows for user-corrected slices to be provided at test-time as input to the refinement model together with the image data to perform a forward pass of the refinement model, thus the "instant" reaction time the authors describe.

Unlike other approaches in the literature, the proposed approach does *not* update the weights of a model using the user-corrected annotations. Presumably this means that the proposed approach requires correction of dense annotations for a number of slices (a set of 12 is selected in the paper) for every new test sample, and the refinement model will not improve its initial performance on subsequent new samples based on annotations of previous samples; i.e., the refinement model weights depend only on the initial small training set. Nonetheless, this idea is novel and the fact that the authors have demonstrated that it improves test-time performance is of interest, both as a practical methodology and as an approach to further explore and evaluate.

In the authors' response regarding annotation time related to correcting dense segmentations of the initial model, the authors state "they [the annotators] are only required to correct small errors and not annotate the segmentation from scratch". This is often no simple task, especially with datasets which involve complex structures such as tumours, where neighbouring regions have subtle differences between them, which a model trained on limited data is likely to get very wrong. The claim that annotators would "only [be] required to correct small errors" would need to be substantiated. Indeed, this is outside of the scope of this paper, but an acknowledgement of how much more time-consuming dense annotation correction can be than the use of scribbles and clicks used in the majority of the literature in this domain would allow readers and practitioners to have a better perspective on the trade-offs of the proposed approach, particularly in terms of time saved on "reaction-time" versus time spent on annotation at test-time.



**Paper Type:**

both

**Questions To Address In The Rebuttal:**

In addition to the questions in the "Detailed Comments" section above, the following should be addressed.

Section "3.2. Slice Acquisition" needs to be significantly improved, either by explaining in the text how the slice selection function works, or by defining all terms and functions in the pseudo-code.

The authors need to more clearly motivate their work, or substantiate the claim that U-net models in interactive segmentation approaches have a high "reaction time" compared to the proposed approach. See first comment in "Weaknesses" section.

It is unclear how the refinement model is updated with the corrected slices. The authors state "The model is trained on the same small training dataset as the presegmentation nnU-Net, but augmented with simulated corrections." Presumably the simulated corrections are applied only on data which were not used for the training of the presegmentation network? Was this data the validation data described in in Section "4.1 Datasets"? Also, are the weights of the refinement model updated with these corrected slices (as done in the cited works such as (Wang et al., 2019b))? Please clarify.

A deep ensemble of nnU-Nets was used to estimate uncertainty for the propose approach. Was a deep ensemble also used for inference, for (a) the presegmentation and (b) the refinement models? Was a deep ensemble also used for inference when compared to test-time augmentation and MC dropout in Section 5.3? If so, the authors should state this and note that a deep ensemble would effectively require more training resources, and also uses several times more parameters as the other approaches.

Was a deep ensemble also used for the P-Net in comparisons to the nnU-Net?

The citation Bredell et al. is not properly referenced in the bibliography, only containing names and title. The authors need to complete the remaining fields, see: https://link.springer.com/chapter/10.1007/978-3-030-00919-9_42

Figure 1 is a bit confusing, specifically where red and green boxes are used in "stage 2" and "stage 3". Firstly, these boxes are not described in the caption or in the text. Secondly, if these boxes are meant to indicate slices of the 3D volume which have been selected for interactive correction, they are somewhat misleading as they appear to highlight different parts of a 2D image, whereas in Section 3.2 the authors described that "a one-shot slice acquisition function selects multiple slice sfor each subject in axial, coronal and sagittal orientation". This implies that entire image slices are selected, and not sub-regions of a slice as suggested by the red and green boxes in Figure 1. Indeed, Figure 2 illustrates "Random Slices" corresponding to what is described in the text.

Typo on line 4 of Algorithm 1: "uncertaintenSlices"

Typo in Section 3.3.:  "are send" should be "are sent"

In Section 3.4, the following sentence is difficult requires clarification: " After training, the model is then able to facilitate the corrected slices during inference to infer a globally refined segmentation from the local corrections."

**Special Issue:**

no

---

### Official Review · Reviewer_7aWU · 2022-01-29

**Confidence:** 4
**Preliminary Rating:** 3
**Recommendation:** Poster

**Summary:**

The authors claim that current interactive 3d segmentation models are limited in scope, since if we require quick reaction to the user input, we cannot use heavy, more accurate models. Their approach to solve this is: instead of presenting the user with the scan from the beginning, they build a presegmentation offline with a nnUnet, find out what are the most uncertain slices automatically, and show the user those slices. The user interacts with them, and a refinement net is used to incorporate the human interactions, inferring a global result from the global presegmentation and the local human corrections. The presegmentation and refinement nets are already trained, so in theory this procedure is fast to the user's eyes.

**Strengths:**

Interactive 3d segmentation methods are of high practical interest. The evaluation on several datasets seems meaningful, rigorous and complete, indicating that the approach works for different data and conditions. The appendices contain interesting information for the reader. Code is to be shared.

**Weaknesses:**

There are several points that I could not understand from the text, the reader gets the feeling that the technical explanation was written very quickly and in a compressed manner so as to fit a page limit, leaving out some important details. I am most confused about the slice acquisition algorithm and the rationale behind the training of the refinement model, please see below my questions.

**Deanonymize Review:**

yes

**Detailed Comments:**

* Q1) The slice acquisition algorithm appears to be key to the proposed approach, yet I don't think the reader can understand it very well from the listing in Algorithm 1. What is SubFromEach? Where in the heuristic are we minimizing correlation between the selected slices? Shouldn't we impose that they are not contiguous slices, somehow, or is this done by the minDist parameter? In this case, what does it have to do with imageSize? How do we select the minUncert parameter, given that the uncertainty returned by the employed techniques is unit-less? I believe this part of the paper deserves a bit more detail (in text), and is not dealt with properly with just a listing of an algorithm. Also, please include the default values of all these parameters in the main text, they are now buried in Appendix B but they appear to be critical to performance.

* Q2)  Shouldn't the training of the refinement network be explained/included in stage 1, rather than in stage 3? It seems to happen before the human enters the process.

* Q3) I think I don't understand Stage 4, or at least Figure 2 and the explanation. From the figure, we select slices of the ground-truth at random, in whichever order, and those get into the net with the scan so that the refinement model figures out how to reconstruct the entire ground-truth volume from a set of random ground-truth slices. I fail to see how this helps in the process of taking a  human interaction with a wrong segmentation and corrects the segmentation.
The explanation in the text is completed with "After training, the model is then able to facilitate the corrected slices during inference to infer a globally refined segmentation from the local corrections." I don't understand this sentence neither, what does "facilitate the corrected slices" mean? Can you please expand your explanation of this point?

* Q4)  Uncertainty is computed at the voxel level, yet the relevant uncertainty here is at the slice level. Did the authors consider aggreggation approaches beyond the simple mean of the per-voxel uncertainties to come up with a per-slice uncertainty score?

* Q5)  For the ECE calculations by the end of the paper, they are indeed highly impacted by class imbalance (the net is likely very confident, and correct, in background voxels all the time, which shrinks the ECE exaggeratedly). May I suggest to produce some kind of per-class ECE? I think something similar has been done here recently (look for Stratified Brier Score):
Orthogonal Ensemble Networks for Biomedical Image Segmentation, AJ Larrazabal, C Martínez, J Dolz, E Ferrante, MICCAI 2021)

#### Minor:
* "The results are the same as the **once** depicted in Figure 3."
* Caption in Fig 6 says a brain is shown, but that does not look like a brain.
* The following reference seems to be a meaningful one to add in this paper, but it was ignored:
iW-Net: an automatic and minimalistic interactive lung nodule segmentation deep network
G Aresta, C Jacobs, T Araújo, A Cunha, I Ramos, B van Ginneken, et al., Scientific reports 9 (1), 1-9, 2019
* I am curious, how come the P-Net model have five references related to it?

**Final Rating After The Rebuttal:**

4: Weak Accept

**Justification Of The Final Rating:**

Well, the authors have clearly worked a lot in order to address my comments and the ones of the other reviewers - the revised paper is half in red font. I see my main concerns answered. Therefore, I would like to 1) Upgrade my decision to Weak Accept, and 2) Wish you luck with reviewer 7mjc :)

**Paper Type:**

both

**Questions To Address In The Rebuttal:**

Mostly, it would be really nice to read some further explanations on the motivation/rationale behind the way the refinement net is trained, I am very confused about that. Further clarifications on the slice acquisition process would also be much welcome. Thanks!

**Special Issue:**

no

---

### Meta-Review · Area_Chair_b86v · 2022-02-20

**Recommendation:** Accept (Poster)
**Confidence:** 5

**Metareview:**

This paper proposes a framework for interactive segmentation. The main contribution is in the use of uncertainty to select slices for user corrections. The paper received mixed scores initially, mainly due to the lack of clarity in important components of the framework. The authors provided very detailed responses, addressing most of the initial concerns and significantly improving the clarity of the paper. This is a case where the discussion phase was very helpful/informative and, eventually, the final scores were two weak-accepts and two borderlines. The reviewers agree that the work has some novelty/merit and could be of practical value (as it improves test-time performance). I concur with this and, even though the technical novelty might be perceived as limited, I believe the practical value of the work deserves publication.

---

### Decision · Program_Chairs · 2022-02-28

Accept